# Origin and Potential Expansion of the Invasive Longan Lanternfly, *Pyrops candelaria* (Hemiptera: Fulgoridae) in Taiwan

**DOI:** 10.3390/biology10070678

**Published:** 2021-07-17

**Authors:** You-Sheng Lin, Jhih-Rong Liao, Shiuh-Feng Shiao, Chiun-Cheng Ko

**Affiliations:** Department of Entomology, National Taiwan University, Taipei City 106332, Taiwan; r09632002@ntu.edu.tw (Y.-S.L.); kocc2501@ntu.edu.tw (C.-C.K.)

**Keywords:** genetic structure, biological invasion, *Pyrops candelaria*, longan lanternfly, habitat suitability

## Abstract

**Simple Summary:**

The longan lanternfly *Pyrops candelaria* (Linnaeus), which feeds on longan trees, has invaded the main island of Taiwan. Thus, this study aimed to (1) trace the origin of invasion, (2) predict habitat suitability and (3) study the risk of lanternfly spread into crop cultivation areas. We reconstructed the longan lanternfly phylogeny to compare haplotypes among different regions. In addition, we predicted the habitat suitability for the species based on MaxEnt and then matched it with longan and pomelo cultivation areas. Our results indicated that the Taiwanese populations of longan lanternfly might originate from the Kinmen Islands and the plain areas of Taiwan are considered to have high habitat suitability.

**Abstract:**

The longan lanternfly *Pyrops candelaria* is a new invasive species on the main island of Taiwan. The introduction of an invasive species may negatively influence the native fauna, flora and environment. Thus, this study aimed to infer the invasion history, predict habitat suitability and potential expansion and assess the risk to crop cultivation areas in Taiwan. Genetic structures of *P. candelaria* from the main island of Taiwan and related regions were analyzed based on partial *COI* and *ND2* sequences. Additionally, machine learning MaxEnt was utilized to study habitat suitability. The results suggested that the Taiwanese populations may originate from the Kinmen Islands and the plain areas of Taiwan are considered to have high habitat suitability. Furthermore, most of the cultivation areas of longan and pomelo crops showed high habitat suitability.

## 1. Introduction

Biological invasions may pose threats to native fauna and flora, the economy and even human health [1], constituting a global economic and ecological threat [2,3,4]. Numerous human activities (e.g., international trade and travel) have enabled certain species to cross their natural geographical barriers to invade a new region [4,5,6]. Understanding the genetic structures of invasive species is helpful for effective management and quarantine [7,8,9]. The mitochondrial cytochrome oxidase subunit I (*COI*) and NADH dehydrogenase subunit 2 (*ND2*) genes have been used in studies of genetic structures (e.g., *Tessaratoma papillosa* (Drury) and *Lycorma delicatula* (White)) [7,8]. Moreover, the maximum entropy method (MaxEnt), a machine learning approach, has been widely used to explore species niches by analyzing species-environment relationships using presence-only data and environmental variables [10]. As only presence data are required, this method has multiple applications, including determining suitable habitats for protected species [10], predicting the distribution of invasive species [11], studying the establishment of invasive natural enemies [12] and modeling distribution shifts caused by climatic changes [13].

Hobbyists aspire to obtain live lanternfly specimens because of their fascinating appearance and scarcity in the market; however, certain lanternfly species could become serious pests. For instance, the spotted lanternfly (*Lycorma delicatula*) is a notoriously invasive species that causes great economic losses to the grape industry [6]. The longan lanternfly *Pyrops candelaria* (Linnaeus) (Hemiptera: Fulgoridae) is a large species with an exaggerated cephalic process and magnificent wing pattern (Figure 1), making it popular among collectors. This species is distributed in China, India, Myanmar, Thailand, Vietnam, Cambodia, Indonesia (Java) and Kinmen, an outlying island of Taiwan (Figure 2) [14,15,16,17]. The major host plants of the longan lanternfly are longan (*Dimocarpus longan* Lour.), pomelo (*Citrus maxima* Merr.) and mango (*Mangifera indica* Linnaeus) [18]. This species was observed for the first time in Wugu and Bali in Northern Taiwan in 2018. On Kinmen Islands, the longan lanternfly is regarded as a mascot. The remarkable appearance renders the species an excellent subject for environmental education. The insect also attracts many wildlife photographers from the main island to Kinmen every year. In addition, the Kinmen Islands are the nearest native region for the longan lanternfly without tight quarantine, since it belongs to the territory of Taiwan. As a result, Kinmen Islands are a possible origin of invasion. Previous studies have reported the longan lanternfly as an agricultural pest [15,19,20], but no significant impact has been observed since its invasion. Nevertheless, the ecological effects of invasive species are unpredictable. Therefore, from the conservation perspective, management of the longan lanternfly should be taken seriously.

This study aimed to trace the invasion history of the longan lanternfly, predict the potential expansion area and assess the risk to crop cultivation areas in Taiwan. First, we hypothesized that the origin of invasion is Kinmen Islands; therefore, genetic structures for the newly introduced populations of *P. candelaria* were reconstructed and the haplotypes among different regions were compared. Second, MaxEnt was used to predict the potential distribution of *P. candelaria* in Taiwan with inferences drawn for habitat suitability under certain climatic conditions, including temperature and precipitation. Finally, the potential distribution areas were matched with information on the host plant cultivation areas. The result is a combination of climatic factors and host plant information to improve the credibility of risk assessment for invasion and potential spread.

## 2. Materials and Methods

### 2.1. Genetic Structure and Phylogenetic Reconstruction of P. candelaria in Taiwan and Related Regions

#### 2.1.1. Sample Preparation, PCR Amplification and Sequencing

A total of 155 *P. candelaria* specimens were collected from the main island of Taiwan (*n* = 105) and several native regions, including China (*n* = 5), Thailand (*n* = 20), Hong Kong (*n* = 3), Macau (*n* = 1), Kinmen (*n* = 20) and Matsu (*n* = 1) (Appendix A). All samples were preserved in 95% ethanol and stored at −20 °C.

Genomic DNA was extracted from the single tarsi of *P. candelaria* specimens using QuickExtract™ DNA Extraction Solution 1.0 (Epicenter) according to the manufacturer’s protocol for insect tissue. Two partial sequences from the mitochondrial cytochrome oxidase subunit I (*COI*) and NADH dehydrogenase 2 (*ND2*) genes were targeted for amplification. The *COI* gene was amplified using the universal primers LCO1490 (5′-GGTCAACAAATCATAAAGATATTGG-3′) and HCO2198 (5′-TAAACTTCAGGGTGACCAAAAAATCA-3′) [21] and the *ND2* gene was amplified using primers ND2-41F (5′-ATGACAATAAGAGTAATAAT-3′) and ND2-819R (5′-TGAAATTAATGATGATCTGA-3′), which were designed in this study. The reaction mixture contained 2 μL of extracted DNA, 0.5 μL of each primer, 4 μL of 5× FIREPol^®^ Master Mix (Solis BioDyne) and 13 μL of ddH_2_O. The PCR included an initial denaturation at 94 °C for 2 min, followed by 35 cycles of 30 s denaturation at 94 °C, 30 s annealing at 47.5 °C and 40 °C for the *COI* and *ND2* genes, respectively, 45 s extension at 72 °C and a final extension at 72 °C for 10 min. Automatic sequencing was performed using an ABI 3730XL DNA Analyzer (Applied Biosystems).

#### 2.1.2. Sequence Analyses and Phylogenetic Reconstruction

Sequences were edited and assembled into contiguous sequences using BioEdit 7.0.5.3 [22] and sequence alignments were performed automatically using the MAFFT 7.0 online server [23]. Haplotype network analysis was applied using TCS 1.21 [24] and modified using tcsBU [25]. Pairwise genetic distances between different populations were estimated using MEGA X with the Kimura 2-parameter model [26].

The best-fit nucleotide substitution model (the GTR + I model) for phylogenetic analysis was estimated based on the Akaike Information Criterion using MrModeltest 2.4 [27]. Phylogenetic analysis was conducted using maximum likelihood (ML) and Bayesian inference (BI). ML analysis was performed using W-IQ-TREE [28] with 1000 bootstrap replicates for validation. BI analysis was performed using MrBayes 3.2 [29]. Two simultaneous Markov chain Monte Carlo simulations were run for 3 × 10^6^ generations with a sampling frequency of 100 generations and the initial 25% of generations were discarded as burn-in. FigTree v1.4.4. [30] was used to edit the consensus tree. *Lycorma delicatula* (White) (accession number: MT079712) was chosen as the outgroup. Two sequences of *P. candelaria* retrieved from GenBank (accession numbers: FJ006724 and KM244702) were added for extra-genetic comparison.

### 2.2. Prediction of Habitat Suitability of P. candelaria

#### 2.2.1. Species Occurrence Dataset

Global occurrence data on *P. candelaria* (Figure 2, Appendix A) were collected from the Global Biodiversity Information Facility [31] and collected in the current study. Repeatedly acquiring data was excluded using Quantum GIS (QGIS) [32] and ensured the presence of only one distribution point in each raster, as shown in Figure 2 to avoid overfitting. In total, 298 localities (Appendix A) were organized using Microsoft Excel for further analysis [33].

#### 2.2.2. Environmental Variables

A total of 19 environmental variables (period: 1970–2000) were collected from WorldClim (https://www.worldclim.org, accessed on 1 April 2021) [34,35] at a spatial resolution of 30 arc-seconds (1 km^2^). These bioclimatic variables were derived from temperature and precipitation (Appendix A), which are considered to be related to the distribution and survival of small arthropods; therefore, they have been widely used in the prediction of species distribution [13,36]. Data were retrieved from the WorldClim database using RStudio (version 2.1) with R language ‘raster’ [37] and ‘rgdal’ [38] packages. These variables were imported into MaxEnt for the initial model to calculate the contribution rate using the jackknife test to avoid multicollinearity [10] and the selected variables are shown in Table 1.

#### 2.2.3. Distribution Modeling

MaxEnt (v.3.4.1) [39] was applied to predict the habitat suitability of *P. candelaria* in Taiwan based on global occurrence data (Figure 2, Appendix A) and environmental variables (Table 1) using the R package ‘dismo’ [40] in RStudio. The performance of diverse models and para (i.e., the feature type and regularization multiplier) in relation to the best fitting model in ten-fold replicates was applied to and evaluated using the R package ‘ENMeval’ [41] and other parameters were set as the default; among the five feature types (i.e., linear, quadratic, product, threshold and hinge), linear, quadratic, hinge and product types were allowed and the regularization multiplier was set to 0.5. Presence-only data were generated pseudo-absences and 10,000 random background points were randomly selected by the MaxEnt model, which either ran 500 iterations of these processes or continued until a convergence threshold of 0.00001 was attained.

The prediction results from MaxEnt modeling were evaluated according to the threshold independent area under curve (AUC) values. Receiver operating characteristic (ROC) curves were used to plot the true-positive rate against the false-positive rate and the AUC was used as a measure of the goodness of fit of the model [10,42]. We selected a test sensitivity of 0% and 10% omission rates (OR) [43,44]. The AUC value ranges from 0 to 1, with higher values indicating higher predictive performance [45]. In the case of default OR, the value at 10% was 0.10, the sensitivity test value at 0% was 0 and poor performance was indicated by a value exceeding the predicted rate [46]. The logistic output was chosen as an estimate of the probability of presence conditioned by environmental variables (i.e., habitat suitability) per grid cell. Jackknifing was used to screen for dominant environmental variables [47]. In addition, we obtained response curves showing the single effects of individual variables on the species occurrence and the curves for all environmental variables (i.e., selected based on the initial modeling).

### 2.3. Comparisons of Habitat Suitability of P. candelaria and Crop Cultivation Areas

Based on the results of a field survey, two economically important host crops of the longan lanternfly, longan and pomelo were selected for comparison. We obtained the 2019 Taiwan crop cultivation areas listed by the Agriculture and Food Administration of the Executive Yuan from the agricultural report resource network database using towns (or districts) as points. The crop cultivation maps were compared with the prediction results of MaxEnt and a histogram of the habitat suitability of crop locations for the longan lanternfly was generated.

## 3. Results

### 3.1. Phylogeny, Haplotype Network and Genetic Distance of P. candelaria in Taiwan and Related Regions

The sequence lengths for the *COI* and *ND2* genes were 658 bp and 548 bp, respectively. The sequences amplified in the present study have been deposited in GenBank under MZ350301–MZ350455 for the *COI* gene and MZ358925–MZ359079 for the *ND2* gene (Appendix A). A total of 31 haplotypes were identified in 157 sequences. Phylogenetic inference using concatenated *COI* and *ND2* genes revealed that all samples from the main island of Taiwan formed the largest clade with 10 samples from Kinmen and one sample from Hong Kong. The second clade comprised one sample from Kinmen and most of the samples from Hainan, thus being the sister group of the largest clade. The phylogenetic tree indicated that the haplotypes from East Asia formed a clade derived from the lineages of Thailand with good support values. The samples from Thailand did not form a monophyletic group but were at the basal position of the ingroup (Figure 3).

In the haplotype network analysis, all the samples from the main island of Taiwan belonged to haplotype H1, sharing the same haplotype with 10 samples from Kinmen and one sample from Hong Kong. These results support the phylogenetic analysis. Haplotype H3 comprised a sample from Macau and partial samples from Kinmen and Hong Kong, being the central haplotype of the network. The other haplotypes contained samples from a single location (Figure 4). The genetic distances varied from 0 to 0.0139 among different populations of *P. candelaria*, with the highest between Hainan (CHN) and Fujian (CFJ). There were no differences among the populations from the main island of Taiwan, including Bali (TBL), Wugu (TWG), Guanyin Mountain (TGM), Beitou (TBT), Linkou (TLK) and Zhonghe (TZH) (Appendix A).

### 3.2. Comparison of Potential Distributions of P. candelaria and Cultivation Areas in Taiwan

Figure 5 shows the OR of prediction for *P. candelaria*. The training set OR (blue line) should be close to the predicted OR (black line). The average OR of the training set for *P. candelaria* was 0.007. The ROC curve output from MaxEnt showed that the AUC of the *P. candelaria* training set was 0.979. According to the standard by which the AUC was evaluated, the prediction model achieved a high performance [46,48]. The potential distribution areas of *P. candelaria* included most of the lowland to low-altitude mountainous areas, but not in mid- and high-altitude ones. In addition, outlying islands (Penghu, Ludao and Lanyu) were considered highly suitable (Figure 6). The host plants longan and pomelo are cultivated in most lowland areas in Taiwan, particularly in western Taiwan, most crop locations had high habitat suitability (Figure 7).

### 3.3. Environmental Variables Influencing the Distribution of P. candelaria

The contribution rate and permutation importance of the environmental variables were determined using a jackknife test (Figure 8, Table 1). In particular, precipitation of warmest quarter (bio18), isothermality (bio03) and precipitation of wettest month (bio13) were the most influential factors for determining habitat suitability for *P. candelaria*, with a total contribution of 93.3%. In contrast, mean temperature of driest quarter (bio09), mean temperature of warmest quarter (bio10), temperature seasonality (bio04) and annual precipitation (bio12) had little influence on the results, with a total contribution of 6.8%. The response curves indicate several patterns of variation (Figure 9). They were generally unimodal, monotonically increasing or decreasing and were not overly complex or suggestive of overfitting. For instance, bio18 was the most important variable for *P. candelaria* and exhibited a monotonically increasing pattern.

## 4. Discussion

In this study, the invasion history and potential expansion of *P. candelaria* were determined. The results of the molecular study revealed that the insect populations of the main island of Taiwan might have originated from the Kinmen Islands. In addition, the comparison of potential distribution and crop locations revealed that *P. candelaria* poses a high risk to both longan and pomelo. Future studies will focus on developing higher resolution molecular markers to assess the expansion pattern of *P. candelaria*. In addition, the habitat suitability results revealed that regular monitoring should be established to assess potential spread.

### 4.1. Invasion History of P. candelaria in Taiwan

This present study investigated the genetic structure of the invasive population of *Pyrops candelaria* in Taiwan and related regions using partial *COI* and *ND2* sequences. According to the results of phylogenetic and haplotype network analysis, all the samples collected from the main island of Taiwan belonged to the same haplotype (H1) with 10 samples from Kinmen and one sample from Hong Kong (Figure 3 and Figure 4), suggesting that both Kinmen and Hong Kong could be possible sources of the invasive populations in Taiwan. From the haplotype network, haplotype H1 was found to be widespread around Kinmen, accounting for 50% of the samples from Kinmen used in this study. For the reason mentioned above, although haplotype H1 also included one sample from Hong Kong, we are more inclined to believe that the invasive population of the longan lanternfly on the main island came from Kinmen. We speculate that tourists traveling to and from Kinmen introduced these insects to the main island.

However, from the phylogenetic analysis, we also noticed that haplotype H1 has a close relationship with haplotypes from China (Figure 3). Therefore, we do not rule out the possibility that haplotype H1 is widespread in East Asia, but this was not fully investigated in this study because of limited sampling. The Thai population occupied a basal position of the tree (Figure 3), implying that this population may be a relatively ancestral lineage compared to East Asian population. Matsu is another outlying island of Taiwan. Whether the sample from Matsu is native is an interesting issue since this species has never been recorded from Matsu before according to Shih et al. [49], and few individuals have been observed in Matsu in recent years. The haplotype from Matsu was unique in the haplotype network (Figure 4), but it could be the result of natural migration from China due to the short distance between the two locations. This question can possibly be resolved after further sampling.

### 4.2. Habitat Suitability, Potential Expansion and Comparison with Crop Cultivation Areas

Invasive species may have adverse effects on native fauna and flora, causing great economic losses [1,2,3,4]. The main island of Taiwan covers an area of approximately 36,000 km^2^ with a maximum elevation of 3952 m and approximately 70% of the land is mountainous and hilly. The lowland areas in Taiwan are mainly concentrated along the western coast [11].

The prediction by MaxEnt revealed high accuracy in both OR and ROC curves, which were used to verify the credibility and accuracy [43,48]. Suitable areas are distributed throughout the plain areas of the main island of Taiwan, except in the mountainous areas (Figure 6). The results indicated that *P. candelaria* posed a high risk of establishing populations in the cultivation areas of both host plants. In addition, there are many wild longan trees in the low-altitude mountains of Taiwan. Based on the habitat suitability of *P. candelaria*, paths of potential spread may take two directions: one may follow the plain to Taoyuan County and another may spread to Yilan County toward the southeast. Therefore, we consider that a pest monitoring mechanism should be established to keep track of potential expansion in areas with a high probability of invasion.

Wu et al. [8] reported that the litchi stink bug, *Tessaratoma papillosa* invaded Taiwan in 2008 and has now spread throughout Taiwan. Because of similar food preferences, the longan lanternfly could also spread throughout Taiwan, which has not happened because of the relatively poor dispersal ability and adaptability of the species. Since 2018, we manually removed *P. candelaria* and it remained confined to Northern Taiwan. This control strategy seemed to be effective in limiting the spread. However, the present study indicated that the longan lanternfly had invaded Taiwan at least once. If natural expansion and artificial introduction occur, this invasive species may become established in crop cultivation areas with high habitat suitability.

### 4.3. Environmental Variables Affect the Establishment of P. candelaria

This study showed that key environmental variables affecting the habitat suitability of *P. candelaria*, namely precipitation of the warmest quarter, isothermality and precipitation of the wettest month, were the most influential factors in the prediction of habitat suitability for *P. candelaria*, with a total contribution of 93.3%. In contrast, mean temperature of the driest quarter, mean temperature of the warmest quarter, temperature seasonality and annual precipitation had little influence on the results, with a total contribution of 6.8% (Table 1). The response curve (Figure 9) of bio18 showed a monotonically increasing pattern, which revealed that summer precipitation positively affected the habitat suitability of *P. candelaria*. However, Wang et al. [15] reported that the occurrence of this species is not related to precipitation. In addition, bio03 showed a monotonically decreasing pattern, which revealed that when the temperature changed significantly, it would decrease the habitat suitability. Wang et al. [15] reported that the occurrence of the species was related to temperature, which lacked laboratory experiments. We considered that the response curve of bio03 revealed that this species preferred a location with warm temperatures and heavy rain during summer.

## 5. Conclusions

Biological invasion can cause economic and ecological damage [2,3,4], and human activity is a key cause of invasions [5]. To the best of our knowledge, this is the first study to report the genetic structure of the invasive *P. candelaria* population in Taiwan. The haplotype from Taiwan is consistent with a widespread haplotype from the Kinmen Islands, which indicates that the invasion of *P. candelaria* may originate from the outlying islands of Taiwan. The prediction of habitat suitability for *P. candelaria* revealed that almost all of Taiwan has high habitat suitability. Under natural conditions, the potential expansion may move in two directions: one follows the plain to Taoyuan County and the other spreads to Yilan County toward the southeast. However, the longan lanternfly could spread to areas with high habitat suitability. Future studies aim to develop molecular markers with higher resolution (e.g., microsatellites) to study the detailed invasion history and expansion pattern of this species. We believe that our study can provide a reference for quarantine policies aimed at outlying islands and for environmental education about exotic species.

## Figures and Tables

**Figure 1 biology-10-00678-f001:**
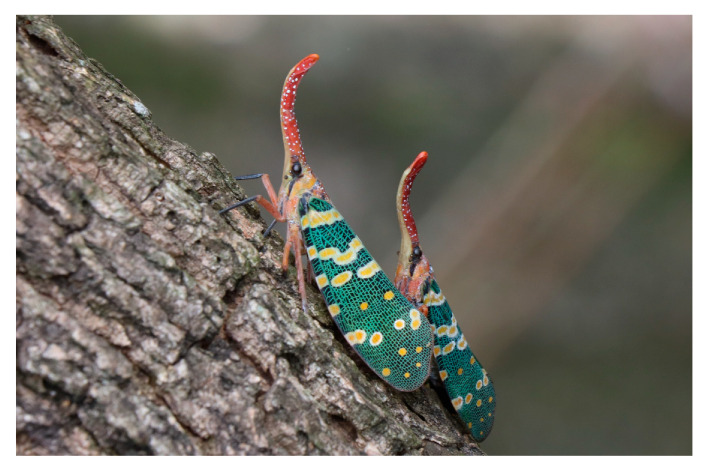
Adult individuals of the longan lanternfly, *Pyrops candelaria* (Linnaeus) on its main host plant, *Dimocarpus longan*.

**Figure 2 biology-10-00678-f002:**
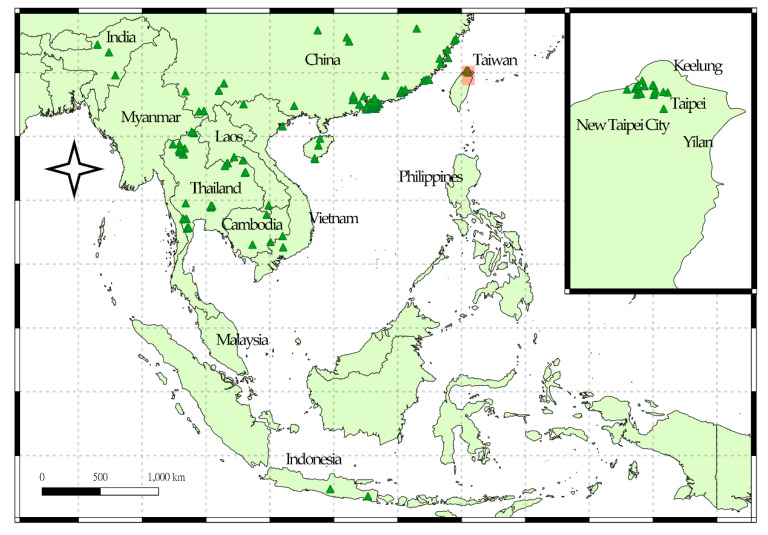
Global occurrence of *Pyrops candelaria*. The triangles represent distribution records.

**Figure 3 biology-10-00678-f003:**
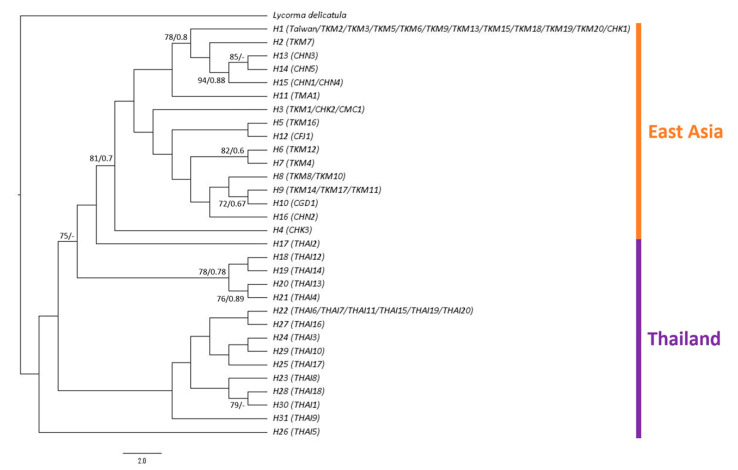
Phylogenetic tree of *Pyrops candelaria* haplotypes based on concatenated mitochondrial *COI* and *ND2* sequences. Numbers at the nodes represent maximum-likelihood bootstrap values and Bayesian posterior probabilities.

**Figure 4 biology-10-00678-f004:**
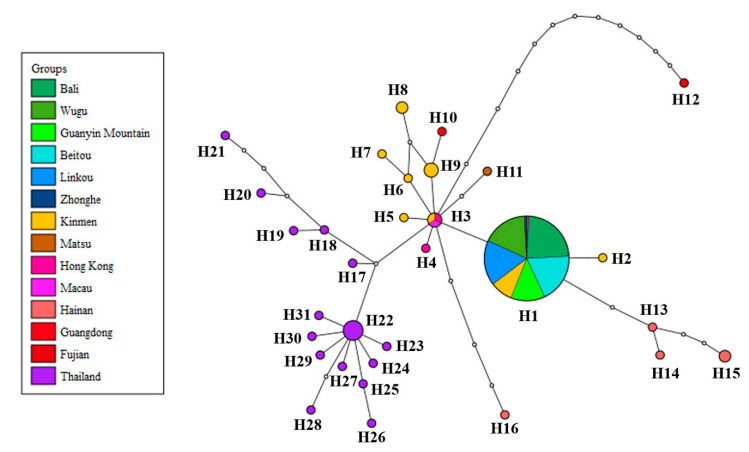
Haplotype network of *Pyrops candelaria* populations based on concatenated mitochondrial *COI* and *ND2* sequences.

**Figure 5 biology-10-00678-f005:**
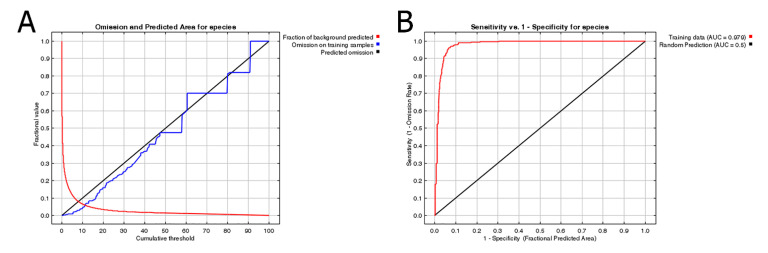
Predictions of habitat suitability for *Pyrops candelaria*: (**A**) Omission rate, (**B**) ROC curve and AUC value.

**Figure 6 biology-10-00678-f006:**
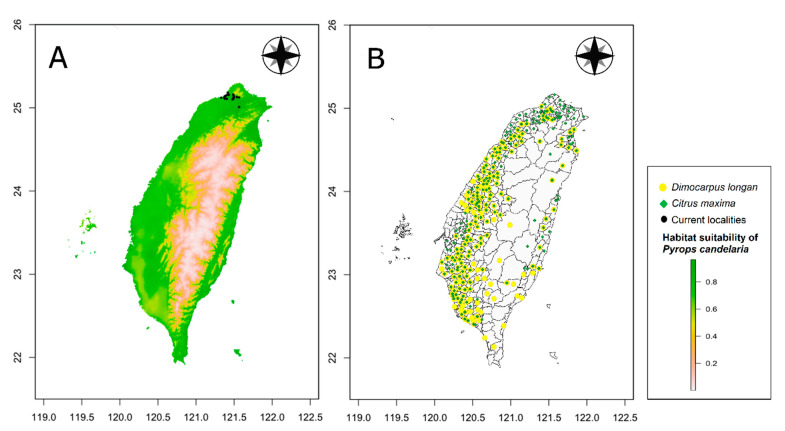
Comparison of potential distribution areas of *Pyrops candelaria* and cultivation areas of *Dimocarpus longan* and *Citrus maxima* in Taiwan: (**A**) Potential distribution, (**B**) Cultivation areas.

**Figure 7 biology-10-00678-f007:**
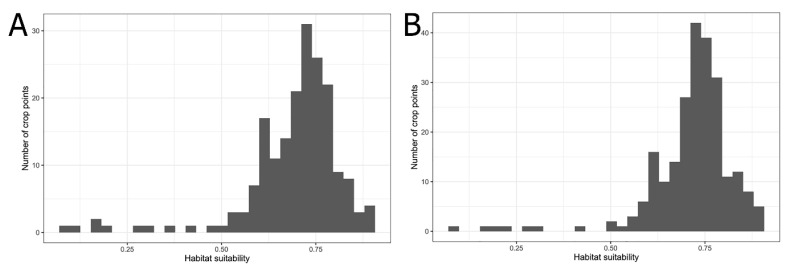
Histogram of the habitat suitability of crop locations: (**A**) Longan; (**B**) Pomelo.

**Figure 8 biology-10-00678-f008:**
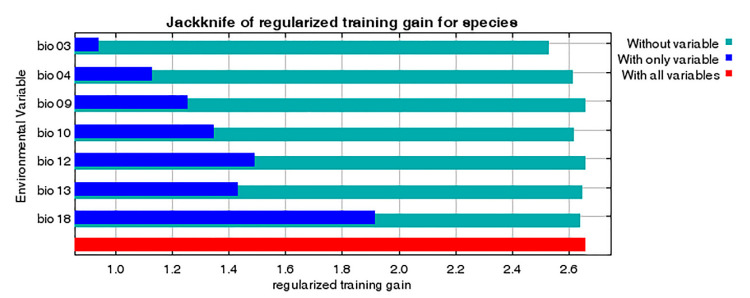
Relative importance of seven selected environmental variables of *Pyrops candelaria* based on the jackknife test.

**Figure 9 biology-10-00678-f009:**
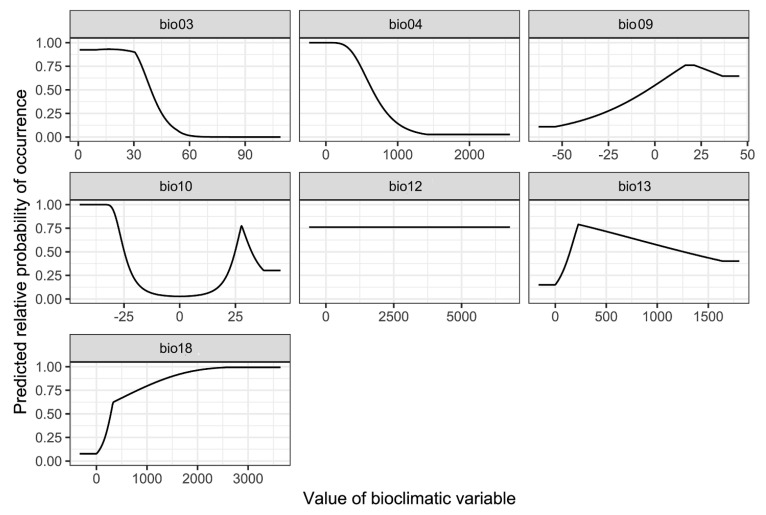
Response curves for seven selected environmental variables of *Pyrops candelaria.*

**Table 1 biology-10-00678-t001:** Contribution rate and permutation importance of selected environmental variables.

Variable Code	Environmental Variables	Contribution Rate %	Permutation Importance %
bio18	Precipitation of warmest quarter (mm)	59.1	5
bio03	Isothermality (°C)	18.5	66.8
bio13	Precipitation of wettest month (mm)	15.7	0.9
bio09	Mean temperature of driest quarter (°C)	4.5	1
bio10	Mean temperature of warmest quarter (°C)	1.9	9.1
bio04	Temperature seasonality (°C)	0.4	17.2
bio12	Annual precipitation (mm)	0	0

## Data Availability

Please see the section “Appendix A”.

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
