# Peer review of "Origin and Potential Expansion of the Invasive Longan Lanternfly, Pyrops candelaria (Hemiptera: Fulgoridae) in Taiwan"

_biology, 2021, doi:10.3390/biology10070678_

Round 1

Reviewer 1 Report

I am concerned by the interpretation of the findings. The authors strongly hint that the introduction is "intentional", without clearly specifying why they think so. This aspect needs re-phrasing because there are no grounds for this conclusion. The authors also need to justify why they think this is not an accident; I found no convincing evidence that this possibility can be excluded.

At one point, the authors mention that the species' appearance makes it a kind of desirable mascot - so a photo of the insect itself may be helpful for the readers.

Author Response

Dear Reviewer

Reviewer 2 Report

please see attachment.

Author Response

Dear Reviewer

Reviewer 3 Report

Here I am providing my evaluation on Lin et al that concerns the putative invasion history, predicted range expansion and potential impact/risk on domestic agriculture (longan and pomelo) of the longan lanternfly that recently was found to introduce in Taiwan. The approach the authors employed to tackle these questions is multidisciplinary as it includes phylogenetic analysis, machine learning-based species distribution modeling and GIS technique. I am no data scientists so I would focus on commenting the phylogenetic analyses and the conclusions derived. 

One major concern is that one of the fundamental hypotheses in this study is built on the statement of local residents alone (L60-63, 4.4), which itself has no scientific or historic foundation, especially for that “hobbyists brought the lanternflies back from Kinmen islands” and “tried to rear them in the agricultural farm as insect pets”. If there is no creditable source of these speculations or citation, I certainly am not convinced this would be a proper hypothesis-testing framework. I will suggest redeveloping the hypothesis.

While the phylogenetic analyses are properly performed, there are several critical issues that the authors may want to address so this article, at least for the phylogenetic component, would be more scientifically-sound. For example, Fig 2 is not clear in terms of phylogenetic structure and relationships among clades as it current form. A major reason is that the authors included all samples from Taiwan (main island) in the tree even though virtually all the lanternflies on the main island share a single identical mtDNA haplotype (in other words, only one individual among those sharing the identical haplotype should be presented in the tree). Plus, readers are able to get a pretty good idea of how hapltypes are related to each other and its association with their geographical origin from the haplotype network.

It’s a bit puzzling that there are two phylogenetic tress (potentially redundant), with the two essentially being the same in terms of phylogenetic structure but in different ways of presentation (especially the figure captions are almost identical). I will suggest deleting Fig. 2 and focusing to develop the discussion on Fig. 4.

Minor comment #1: where are Bali, Wugu etc? There is no geographical reference for these sites. Without such information, readers would have a hard time wrapping their heads around where these sites are distributed geographically. Indicating these sites on Fig. 1 would fix it. 

Minor comment #2: did the authors concatenate sequences of the two genes when they run the phylogenetic analyses? No information is available in the current version of the manuscript.

Minor comment #3: which nucleotide model is the best fit? Th authors mentioned such test in the section M&M (L109-110) but the result is absent in the section Result.

Minor comment #4: Consider removing Table 2 or moving it to supplementary files as Table 2 does not seem to provide more information than the other phylogenetic analyses. Also, again all main island samples share a single haplotype….so please delete L288-289 (the pairwise genetic…) - the statement here is redundant as all these sample share an single haplotype (so there is no genetic distance, ins't it?). 

Minor comment #5: Please explain at least in a supplementary Table what those sample names represent

Minor comment #6: L296-297 the wording is not phylogenetically professional enough - the best the authors can claim from the data is that “the Thai population occupied a basal position of the tree, implying that this population may be a relatively ancestral lineage compared to East Asian population….”

Minor comment #7: L304-305 successfully invaded Taiwan only once? How do you differentiate this scenario from another in which individuals bearing the same haplotype get introduced repeatedly, which is considered more than one successful invasion? I suggest rewording this statement.

Minor comment #8: pregnant female should be “gravid” female

Minor comment #9: L305-314, the entire paragraph is way too speculative and was not supported by any line of your data. It would make more sense that this paragraph is removed from the text or at least a significant revision effort is needed.

Minor comment #10: L298 where is Matsu? How is the population a point that deserves to be mentioned in the discussion? Any geographical significance (e.g., historical population expansion)? Please indicate the significance and add more background info if discussion on this population is desirable.

Minor comment #11: 4.4 the entire paragraph is a deviation from the main focus of this study and gains almost no support from the data itself. I suggest moving this paragraph to the section Intro as it introductory nature.

Author Response

Dear Reviewer

Round 2

Reviewer 1 Report

Thank you for the careful revision. Minor linguistic improvements are still needed, pse see the enclosed revision.

Author Response

Dear Reviewer

Reviewer 3 Report

I thank the authors for taking into account my comments and for taking the time to address my questions. Although the manuscript has been largely improved, I am afraid I cannot recommend to accept for publication until the authors properly justify their hypothesis on the putative invasion source. The authors claimed that the lanternfly was introduced to the main island due to hobbyist's desires (any citation or scientific support on this?), and that the field populations are the results of individuals escaping from some agricultural farms in which these hobbyists attempted to rear them (again I would love to see some citation and scientific support to this notion)? It would be scientifically inappropriate to build a core hypothesis on something without scientific basis such as rumor or gossip. I suggest to drop this wording. In fact, the authors have a pretty solid justification in the revised version: Kinmen is the nearest native range for the lanternfly without quarantine, and I definitely encourage the authors to develop more of the hypothesis based on this, instead of the interviews with local resident.

Author Response

Dear Reviewer
